# What Else Can Be Done by the Spinal Cord? A Review on the Effectiveness of Transpinal Direct Current Stimulation (tsDCS) in Stroke Recovery

**DOI:** 10.3390/ijms241210173

**Published:** 2023-06-15

**Authors:** Paola Marangolo, Simona Vasta, Alessio Manfredini, Carlo Caltagirone

**Affiliations:** 1Department of Humanities Studies, University Federico II, 80133 Naples, Italy; ale.manfredini@studenti.unina.it; 2Department of Psychology, Sapienza University of Rome, 00185 Rome, Italy; simona.vasta@uniroma1.it; 3IRCCS Fondazione Santa Lucia, 00179 Rome, Italy; c.caltagirone@hsantalucia.it

**Keywords:** transpinal stimulation, transcutaneous spinal stimulation, neuromodulation, spinal cord, post-stroke, brain injury, neurorehabilitation

## Abstract

Since the spinal cord has traditionally been considered a bundle of long fibers connecting the brain to all parts of the body, the study of its role has long been limited to peripheral sensory and motor control. However, in recent years, new studies have challenged this view pointing to the spinal cord’s involvement not only in the acquisition and maintenance of new motor skills but also in the modulation of motor and cognitive functions dependent on cortical motor regions. Indeed, several reports to date, which have combined neurophysiological techniques with transpinal direct current stimulation (tsDCS), have shown that tsDCS is effective in promoting local and cortical neuroplasticity changes in animals and humans through the activation of ascending corticospinal pathways that modulate the sensorimotor cortical networks. The aim of this paper is first to report the most prominent tsDCS studies on neuroplasticity and its influence at the cortical level. Then, a comprehensive review of tsDCS literature on motor improvement in animals and healthy subjects and on motor and cognitive recovery in post-stroke populations is presented. We believe that these findings might have an important impact in the future making tsDCS a potential suitable adjunctive approach for post-stroke recovery.

## 1. Introduction

Often referred to as the “Cinderella of the Nervous System” in the literature, for several decades, the spinal cord was just considered a bundle of nerves connecting the brain to the body [1]. Many studies conducted on the spinal cord in the twentieth century were fostered by the dramatic consequences of spinal cord injury (SCI), a devastating neurological condition caused by sudden trauma to the spinal cord resulting in severely impaired sensorimotor functions, dramatically affecting an individual’s independence and his/her social and psychological status [2]. Hence, several investigations were carried out on animal models and humans to highlight activity-dependent spinal cord plasticity with skill acquisition early in development and maintenance later in life [1,3,4]. Vahdat et al. [5] were the first to show local learning-induced plasticity in the human spinal cord during motor learning by simultaneously acquiring functional magnetic resonance imaging (fMRI) of the brain and of the spinal cord. Indeed, they found learning-related activity in the cervical spinal region (C6–C8), which was independent of cortical sensorimotor structures’ activation. As suggested by the authors, these data indicate that the spinal cord acts as an active functional component of the human motor learning network, thus, contributing to the learning process [5]. Indeed, recent research by Ocklenburg et al. [6] revealed molecular mechanisms for epigenetic regulation within the spinal cord, which, in turn, might establish the development of handedness in humans. Hence, the hemispheric asymmetries in the human brain may be influenced by gene expression asymmetries in the spinal cord segments innervating the hands and the arms [6]. More recently, Weiler and colleagues [7] carried out a series of experiments testing the efficiency of the spinal feedback pathways for correcting movements. Their results showed that the neural spinal fibers that regulate triceps and biceps muscle reflexes are not only involved in stretching muscle fibers, but they can also integrate sensory information from the arm in order to support postural control of the hand, therefore, implementing sophisticated hand control. All these findings have, therefore, motivated researchers to further investigate plasticity changes in the spinal cord, and, thus, it has since become a target of interest for non-invasive brain stimulation (NIBS) techniques.

Among the NIBS, transcranial direct current stimulation (tDCS) requires the application of a weak electrical current (1 or 2 mA) over the scalp, through two electrodes (most commonly 5 × 5 cm or 5 × 7 cm) with the active electrode placed over the target area and the reference electrode over the contralateral orbit or the shoulder [8,9]. It is generally agreed that anodal tDCS (A-tDCS) leads to an increase in cortical excitability, while cathodal tDCS (C-tDCS) determines an inhibitory effect [10].

More recently, to exploit spinal cord plasticity and learning capacities, transpinal direct current stimulation (tsDCS) has also been applied. In a similar fashion to tDCS, tsDCS also delivers a weak electrical current (2 mA) through two electrodes with the active electrode placed over the spinal vertebrae (usually the thoracic ones but see [11]) and the reference electrode over the shoulder [12,13]. The mechanisms of action of tsDCS have been mainly studied in animals suggesting that tsDCS acts locally at the spinal level and on ascending/descending spinal pathways [14,15,16]. Indeed, while anodal tsDCS increases motor response latency in mice, cathodal tsDCS enhances spinal circuits’ excitability [14,15]. Ahmed and Wieraszko [15] also studied whether tsDCS effects in mice partly depend on glutamate-analogue aspartate release, and they found that cathodal tsDCS increased aspartate, whereas anodal tsDCS reduced it. More recently, Samaddar and colleagues [17] reported tsDCS effects on newly generated spinal cells in mice. Both cathodal and anodal tsDCS increased the expression of brain-derived neurotrophic factor (BDNF), which, in turn, stimulated newly generated cells. Similar effects were found in humans [18,19] revealing that anodal tsDCS influences BDNF production; thus, it impacts spinal plasticity and may influence spontaneous recovery after spinal cord injury or disease. Together with these findings, some other studies in humans have shown that tsDCS exerts its effects at the cortical level by using neurophysiological techniques.

Schweizer et al. [20] investigated tsDCS supraspinal effects by using resting state functional magnetic resonance (rs-fMRI). In a double-blind, crossover study, rs-functional connectivity was measured before and after anodal, cathodal, and sham tsDCS (20 min, 2.5 mA, active electrode centered over T11, and reference electrode over the left shoulder) in twenty healthy participants. Compared with sham, both anodal and cathodal tsDCS exerted connectivity changes in the primary sensory area, in the insula and in the thalamus (see also [21,22]). Several promising studies have also evaluated the effects of tsDCS on the motor cortex by using motor evoked potential (MEP) elicited through transcranial magnetic stimulation (TMS) applied over the motor cortex [23,24,25]. Results showed that cathodal thoracic tsDCS enhances motor values, while sham and anodal polarization have no significant effects [23,24,25]. The hypothesis was advanced that cathodal tsDCS improves motor unit recruitment due to GABAergic inhibition and post-synaptic overexcitation [23]. More recently, Knikou et al. [26,27] evaluated changes in spinal motor neuron excitability by coupling TMS delivered over the motor cortex with tsDCS delivered over the thoracic-lumbar vertebrae (T10 to L2). The results revealed generalized depression in transpinal evoked potentials’ (TEPs) recruitment from the ankle muscles of both legs. The authors concluded that this combined approach could modulate cortical and spinal synaptic activity together [26,27] (see also [28]).

In summary, considering the studies to date, a large amount of evidence suggests that tsDCS is able to induce local and cortical neuroplastic changes and to exert activation in cortical and corticospinal pathways in humans. It has also been shown that thoracic tsDCS exerts its influence on interhemispheric unbalance, which is a frequent consequence of stroke [25]. Thus, tsDCS, through its supraspinal effects, might be suitable for motor and cognitive recovery in post-stroke individuals.

The aim of this review is to provide a comprehensive evaluation of the literature related to tsDCS effects on motor improvement in animals and healthy subjects and on motor and cognitive functions in post-stroke individuals. Indeed, the effectiveness of tsDCS in exerting neuroplastic changes at the cortical levels, particularly into the sensorimotor areas, has motivated some researchers to investigate its use as an additional strategy for the improvement of motor and cognitive disorders in post-stroke populations. With regard to cognitive recovery, the hypothesis has been advanced that tsDCS, by influencing activity in the sensorimotor network, would be efficacious for enhancing those aspects of language related to motor schemata, such as action verbs and speech articulation [29,30,31,32,33,34,35].

## 2. Search Strategy and Selection Criteria

We conducted this study using the scope reviews’ methodological framework. We searched for tsDCS articles on motor improvement in animals and healthy individuals and on motor and cognitive recovery in post-stroke populations on two databases, PubMed and Scopus, and other sources. Three different searches were conducted using different keywords combined with the Boolean operator “AND” and “OR”. The search period was set from January 2015 to April 2023. Keywords included: (1) (“tsDCS” OR “transpinal stimulation” OR “transcutaneous spinal stimulation”) AND (“Motor”); (2) (“tsDCS” OR “transpinal stimulation” OR “transcutaneous spinal stimulation”) AND (“Stroke”); (3) (“tsDCS” OR “transpinal stimulation” OR “transcutaneous spinal stimulation”) AND (“Cognition”). Included articles met the following criteria: (i) only studies applying transpinal direct current stimulation; (ii) only studies on motor improvement in animals and healthy subjects; (iii) only studies on motor recovery in post-stroke populations; and (iv) only studies on cognitive recovery in post-stroke populations.

Studies in which tsDCS was applied in SCI individuals or for chronic pain diseases were not included as most of these studies referred to neurological diseases other than stroke (i.e., traumatic brain injury; multiple sclerosis; and polyneuropathy, see for example [36,37]. After eliminating duplicates, all potentially relevant full texts were screened by the authors (AM and SV) independently of one another to exclude non-eligible items.

## 3. Data Extraction and Analysis

A total of 269 articles were retrieved through Pubmed and Scopus database searching, and 6 articles through other sources; hence, the total number of identified articles was 275. After the removal of 134 duplicates, a total of 141 articles remained out of which 106 articles were excluded because the title or abstract did not deal with the review research topic, and 6 were removed because they referred to reviews. A total of 29 articles were considered eligible for full-text screening. After full-text screening, another 4 articles were removed since they were not related to the review topic (see Figure 1). The selected 25 articles were rearranged into four subgroups according to the two principal aims of the review: (1) studies on tsDCS effects for motor improvement in animals (N = 7) and (2) in healthy subjects (N = 10); (3) studies on tsDCS effects for motor (N = 5) and (4) cognitive (N = 3) recovery in post-stroke populations (see Figure 2 and Table 1, Table 2 and Table 3).

The results obtained in this review are shown in Table 1 and Table 2 for tsDCS studies on motor improvement, respectively, in animals and healthy subjects and in Table 3 for tsDCS studies on motor and cognitive recovery in post-stroke populations. As reported in Table 1, we identified the effects of tsDCS on motor improvement in animals in seven out of seventeen studies. Some studies showed increased motor neuron responses after cathodal tsDCS and decreased responses after anodal tsDCS applied over the thoracic-lumbar [38] and cervical regions [44] while some others showed a facilitation of motoneuron responses after anodal tsDCS, while cathodal effects were not significant [39,40,41,42]. In the Song and Martin study [43], both cathodal and anodal tsDCS immediately increased spontaneous motor unit firing during stimulation, but the administration of the L-type calcium channel blocker Nimodipine decreased motor responses only after cathodal tsDCS.

As shown in Table 2, the effects of tsDCS on motor improvement in humans were identified in 10 out of 17 studies. Most studies targeted the thoracic vertebrae (T-11-T12; [45,46,47,48,49,50,52,53] (but see [51,54]) showing that anodal tsDCS can be effective in preventing fatigue and in enhancing different whole-body movements [45,48]. It can also improve sleep and restless leg symptoms (RLSs) in idiopathic RLS subjects [50]. In Clark et al.’s study [52], thirty minutes of anodal tsDCS combined with a complex terrain motor task determined larger and more consistent retention of performance than the sham condition.

Some other studies reported larger effects for cathodal than for anodal tsDCS in sprint performance [46] and an increase in MEP’s amplitude after cathodal tsDCS combined with a treadmill exercise compared to the sham condition [47]. Conversely, anodal tsDCS reduced MEP’s amplitude [47]. Similarly, Yamaguchi et al. [49] found that cathodal tsDCS can facilitate voluntary motor output. Beneficial effects have also been reported for anodal tDCS over the motor cortex combined with cathodal tsDCS in the mean errors scores and in the reaction times of a selective attention task in experienced boxers and taekwondo practitioners [51,54]. Only Fava De Lima et al. [53] did not report significant changes in the postural sway of seventeen healthy individuals during quiet standing after anodal or cathodal tsDCS.

As reported in Table 3, the effects of tsDCS on motor recovery were identified in five out of eight post-stroke studies. In the three studies by Picelli et al. [55,56,57], different groups of chronic stroke patients with mild to severe residual walking impairment received 10–20 min of robot-assisted gait training sessions, five days a week, for two consecutive weeks. In two works, anodal tDCS was applied over the ipsilesional primary motor cortex and the reference electrode above the contralateral orbit while the tsDCS cathode was positioned over the thoracic vertebrae (T9-T11) and the anode above the contralateral shoulder [55,56]. Results showed an improvement in walking abilities, and the effects persisted at the two-week follow-up. Similar effects were shown using the same tsDCS montage by applying the tDCS cathode over the cerebellum [57]. In the Paget-Blanc et al. study [58], anodal tsDCS over the cervical vertebrae combined with peripheral nerve direct current stimulation decreased upper limb spasticity in stroke patients. These results persisted for five weeks after treatment and were accompanied by improved motor function. In this group of studies, only Awosika et al. [59] did not find significant differences in walking speed capacity after anodal tsDCS over the thoracic vertebrae (T11) in a group of thirty chronic post-stroke patients.

As shown in Table 3, to date, only three studies from the same research group [33,34,35] have investigated tsDCS effectiveness for cognitive recovery in post-stroke patients. Marangolo et al. [33] applied anodal, cathodal, and sham tsDCS over the thoracic vertebrae (T10-T11) in three different experimental sessions in fourteen left hemisphere post-stroke patients. tsDCS was delivered for 20 min (2 mA) while the patients concomitantly performed a verb- and noun-naming task. Each experimental condition was run in five consecutive daily sessions over three weeks. After anodal tsDCS, all patients exhibited a greater improvement in verb naming compared to the other two conditions. This improvement was still present after one week from the end of the treatment. On the contrary, the amount of improvement found in the noun-naming task did not differ among the real and the sham conditions. Similar results were obtained, in a different group of sixteen left hemisphere post-stroke patients, in a recent rsfMRI study [34]. In this study, connectivity changes were present in a cerebellar–cortical network involving action-related cortical regions such as the left cerebellum, the right parietal, and the premotor cortex [34]. In Pisano et al.’s study [35], after five days of anodal tsDCS combined with a repetition task for articulatory deficits, ten left-brain-damaged patients exhibited better accuracy in repeating the treated items compared to the sham condition. These results persisted at one week follow-up and increased performance in other oral language tasks (i.e., picture description, noun and verb naming, word repetition, and reading).

## 4. Discussion

This review aims to present a comprehensive analysis of the literature related to the effects of tsDCS on motor improvement in animals and healthy individuals and on motor and cognitive recovery in post-stroke populations.

As reported in the Introduction, research on spinal cord potentiality for neurorehabilitation began in recent years when animal studies revealed plasticity effects at the spinal cord level induced by tsDCS [14,15,16]. Ahmed and colleagues [14,15,16] were the first to investigate tsDCS modulation effects on the activity of spinal motor neurons and corticospinal transmission in mice. They found that while anodal tsDCS increased latency of the tibial nerve motor response evoked at the cortical level, cathodal tsDCS enhanced muscle contractions. The hypothesis was advanced that changes in neuronal excitability partly depended on an augmented concentration of glutamate, the major excitatory neurotransmitter in the spinal cord [15]; cathodal tsDCS increased its concentration, whereas anodal tsDCS reduced it. Ahmed et al. [16] also hypothesized that during cathodal tsDCS the glutamate increase was accompanied by an enhancement in GABA receptor blockers, which further augmented spinal circuit excitability. It has also been reported that the susceptibility of tsDCS on spinal plasticity is significantly influenced by BDNF [17,18], a key mediator for synaptic plasticity, neuronal connectivity, and dendritic arborization [60].

Since glutamate activates both ionic and G-protein coupled receptors, which are responsible for a lasting increase in efficacy of synaptic transmission in the spinal cord [61] and in the brain [62] (i.e., long term potentiation (LTP)), the hypothesis was advanced that tsDCS might be used as an additional technique to enhance motor activity [45,46,47,48,49,50,51,52,53,54] and to facilitate motor and cognitive recovery in post-stroke populations [33,34,35,55,56,57,58,59,63].

Indeed, according to the earliest reports [14,15,16], all the results in animal studies confirmed that both anodal and cathodal tsDCS induce local and cortical plastic changes [38,39,40,41,42,43] in rat motoneurons, which considerably outlast the time of polarization [38,39,40,41,42]. Most studies found a depolarization and, thus, an increase of motor responses following cathodal tsDCS, and a hyperpolarization and, thus, a decrease of motor responses following anodal tsDCS [38,39,40,41,42,43,44]. In the authors’ hypothesis, cathodal tsDCS increases evoked synaptic transmission, and, by depolarizing cell bodies, it augments motoneuron activity through induction of calcium release [14,15,16,38], while anodal tsDCS exerts opposite effects [43,44]. Fewer studies found facilitatory effects after anodal tsDCS [39,40,41,42] calling into question the distinction between motoneuron firing induced by intracellular stimulation and the firing induced by synaptic activation. Indeed, only in the first case, the anodic current depolarizes the cell membrane since it directly influences the cell body bypassing the synaptic system [39,40,41,42].

With regard to the effects of tsDCS on motor improvement in humans, some studies showed that anodal tsDCS enhances locomotor skills and reduces fatigue [45,48,52] (but see [53]), and it can also improve sleep and restless leg symptoms (RLSs) in idiopathic RLS subjects [50]. It was suggested that anodal tsDCS interferes with the regulatory actions of the spinal circuits on motor unit excitability by creating a permissive state in spinal circuits where a lasting resistance to central fatigue is enhanced, ameliorating motor performance [45]. It was also proposed that anodal tsDCS facilitates the acquisition and retention of locomotor skills by prolonging downregulation of the H reflex on local spinal circuits or by modulating alpha motoneuron excitability [48]. Using resting-state fMRI, Zeng et al. [50] showed that anodal tsDCS normalized the activity in the right anterior insula and temporal cortex of RLS patients. Since these structures are involved in emotional processing, in the authors’ hypothesis, anodal tsDCS contributed to sleep and emotional recovery in these patients by reducing their pain sensitivity [50].

Some other studies reported facilitation effects for cathodal tsDCS in locomotor activity [46,47,49]. However, both in the Sasada et al. and Yamaguchi et al. studies [46,49], cathodal tsDCS over the lumbar [46] or the thoracic region [49] only transiently facilitated peak acceleration in a cycling [46] and in a ballistic motor task [49]. Thus, in contrast to animal studies, tsDCS effects applied for 10 to 15 min disappeared after few minutes [46,47,48,49]. The authors speculated that, different from what happens with animals, which, as anesthetized, have little neural activity during tsDCS experiments, the neural state in awake humans probably only transiently influences the depolarization exerted by cathodal tsDCS over the spinal sensory pathways [46]. In Albuquerque et al.’s study [47], cathodal tsDCS over the thoracic region was combined with a single session of repetitive TMS (rTMS) over the motor cortex during a treadmill walking exercise. They found an increase in cortical and spinal excitability measured using MEP and the nociceptive flexion reflex (NFR), respectively. The authors hypothesized that while tsDCS acted locally over the spinal circuits, rTMS probably promoted changes in the presynaptic inhibition of spinal motoneurons [47]. Interestingly, beneficial effects were found in athletic performance of experienced boxers and taekwondo practitioners also by combining anodal tDCS over the motor cortex with cathodal tsDCS over the thoracic region [51,54]. Indeed, the simultaneous stimulation of motor cortex and spinal cord significantly improved selective attention and reaction times, which, in turn, enhanced athletic performance. In the authors’ hypothesis, the combined effects of tDCS and tsDCS might be attributed to an alteration of spontaneous neural activity and membrane potentials of the cortical and corticomotoneuronal cells, respectively [51,54].

Thus, although studies have turned up mixed results, all these findings nevertheless suggest that, through modulation of the sensorimotor spinal pathways, tsDCS exerts its influence over the sensorimotor cortex by facilitating motor performance. This evidence has given rise to the possibility of investigating tsDCS application as a therapeutic tool in motor recovery after post-stroke injury.

Picelli et al. [55,56,57] were the first to show that multiple sessions of cathodal tsDCS lead to persisting improvement in walking in chronic post-stroke patients. Similar to previous studies [51,54], cathodal tsDCS was delivered simultaneously with either anodal tDCS [55] or cerebellar DCS [56,57]. In their first study [55], the authors hypothesized that tDCS over the ipsilesional lower limb area of the primary motor cortex reduces asymmetry in the transcallosal inhibitory drive, thus improving the strength of the affected lower limb. In parallel, cathodal thoracic tsDCS has improved motor unit recruitment due to local GABAergic system inhibition and glutamate excitatory effects [14,15,16]. With regard to the two studies, which combined cerebellar DCS with cathodal tsDCS, the authors pointed to the role of the cerebellar (dentate nucleus) thalamo–cortical pathway in exerting inhibition over the contralateral motor cortex through the activity of the Purkinje cells. This, as also proposed in their first study [55], reduced interhemispheric unbalance favoring an increase in the affected lower limb performance [56,57].

Among this group of studies, the approach used by Paget-Blanc et al. [58] represents an innovation that combines tsDCS with peripheral nerve direct current stimulation to improve muscle spasticity in post-stroke patients. This simultaneously targets the spinal cord and the peripheral nerves boosting the electrical stimulation effects to decrease spasticity. In this context, it is also worth noting that the enhancement of corticospinal network plasticity to increase synaptic connectivity for motor rehabilitation after stroke has also been shown by using vagus nerve stimulation (VNS) in conjunction with rehabilitative training in humans and animals [64,65].

It seems therefore likely that, to exert an effect over time on motor recovery following a stroke, either cathodal or anodal tsDCS should be combined with cortical [55,56,57] or peripheral stimulation [58]. Accordingly, in the Awosika et al. study [59] anodal tsDCS alone did not improve walking speed and capacity in thirty chronic post-stroke patients.

In light of these results, some studies have also recently investigated the potential of tsDCS for language recovery in left hemisphere post-stroke patients [33,34,35]. Indeed, since the semantic characteristics of action verbs also contain the sensorimotor features to perform the action [30,31,32], this implies that the motor schemata of action verbs are partly represented in the sensorimotor regions. Thus, given that tsDCS exerts its influence also at the cortical level and, particularly, in the sensorimotor region [20,21,22,23,24,25,26,27], Marangolo et al. [33,34] hypothesized that tsDCS would contribute to the recovery of action verbs (i.e., to swim). In both studies, the results confirmed the authors’ hypothesis on two different groups of left hemisphere post-stroke patients [33,34]. Indeed, a significant greater improvement in verb naming was found only after anodal tsDCS compared to the cathodal and sham conditions [33,34]. Interestingly, in Marangolo et al.’s study [34], the improvement found in verb naming positively correlated with connectivity changes, measured using rs-fMRI, in a cerebellar–cortical network recruiting motor regions known to be involved in action-related verb processing. Thus, tsDCS exerts its influence at the cortical level also in language tasks [33,34]. More recently, the effectiveness of tsDCS in the recovery of language was confirmed by Pisano et al. [35] in a study on speech articulation disorders due to left hemisphere post-stroke injury. Indeed, as with action verbs, the ability to articulate words is related to motor schemata; thus, it partly involves the activation of the motor cortex [29,66]. Pisano et al. [35] found that all patients benefited from anodal tDCS combined with a repetition task, thus improving their ability to articulate words. In all of the above studies, the hypothesis was advanced that anodal tsDCS decreased activity of the sensorimotor inhibitory interneurons at the cortical level [67] through inhibition of the ascending spinal pathway [24], which, in turn, improved the efficacy of the sensorimotor regions and, thus, speech articulation.

In summary, all these results provide evidence of tsDCS supraspinal effects suggesting that it may be promoted as a non-invasive intervention to target cortical sensorimotor networks in post-stroke recovery. It is important to highlight that the therapeutic application of tsDCS has also recently moved forward into clinical conditions other than stroke. A study by Pisano et al. [68], for instance, has suggested that tsDCS combined with cognitive training is efficacious for improving motor planning abilities in Alzheimer’s patients, while Benussi et al. [69] have shown that cathodal tsDCS combined with anodal cerebellar tDCS improves motor and cognitive functions in patients with neurodegenerative ataxia.

In conclusion, although tsDCS’s place in the therapeutic armamentarium remains to be determined, the hypothesis might be advanced that tsDCS will represent a promising therapeutic adjunctive approach in neurorehabilitation. Indeed, the heterogeneity of the protocols and electrode montages used thus far does not allow us to establish, to date, which training is more or less effective to reach persistent tsDCS effects over time. It seems likely that a simple increase of neuronal excitability at the spinal or cortical level through tsDCS is well suited to enhance motor performance in tasks that rely on muscle force [45,49]. However, it remains to be determined if this effect is more easily achieved by applying anodal [45,48,52] or cathodal tsDCS [46,49] since both polarities, through different mechanisms, seem to exert positive effects on motor performance [45,46,48,49].

Some studies have also suggested combining tsDCS with tDCS over the brain or the cerebellum for enhancing athletic performance [51,52,53,54] and motor recovery in post-stroke patients [55,56,57]. Based on the earlier tDCS or tsDCS findings on motor performance, in which either of such modalities was found to effectively enhance motor outcomes [70,71,72], the efficacy of this combination approach is not surprising since it simultaneously acts at the peripheral and cortical levels reinforcing the potential of a single stimulation rehabilitation protocol. Nevertheless, the question of whether each modality on its own is more effective or whether the combination of the two techniques is additive remains to be determined and might represent an interesting challenge for future studies. It should also be noted that in some of these studies [55,56,57], the significant differences in motor performance at the first post-treatment evaluation were not maintained either at the two- or at four-week follow-ups. Thus, a further issue that merits investigation in the future is to understand which protocol guarantees the longest tsDCS effects.

Indeed, because of tsDCS’s ability to modulate cortical excitability and to induce beneficial neuroplastic changes, we believe that this technique opens new perspectives in the treatment of motor and cognitive disorders in post-stroke patients.

Two further tsDCS advantages are also worth considering. Its first advantage is its ease of application. Due to the absence of hair on the spinal cord, tsDCS electrodes can be more easily applied over the skin compared to tDCS ones. This guarantees good contact over the skin with a reduction of impedance levels. Accordingly, high impedance is an indicator of poor conductivity and may be the result of poor electrode setup, as with inadequate parting of the hair over the scalp [73]. Secondly, since intervertebral disks have higher electrical conductivity compared to spongy bones [74,75], this means that the electrical current applied over the spinal cord more rapidly reaches the nervous fibers than the same amount of current applied through tDCS over the scalp. In fact, the application of tDCS over the scalp may reduce the current intensity due to the spongy structure of the skull’s bones [74]. Therefore, we might suggest that different than previous tDCS studies (see for review [76]), tsDCS would be effective for the recovery of cognitive functions related to sensorimotor processing, because the application of the electrode over the spinal cord would simultaneously influence different parts of the sensorimotor network. This would prevent the need to decide which part of the sensorimotor system should be targeted with tDCS.

## Figures and Tables

**Figure 1 ijms-24-10173-f001:**
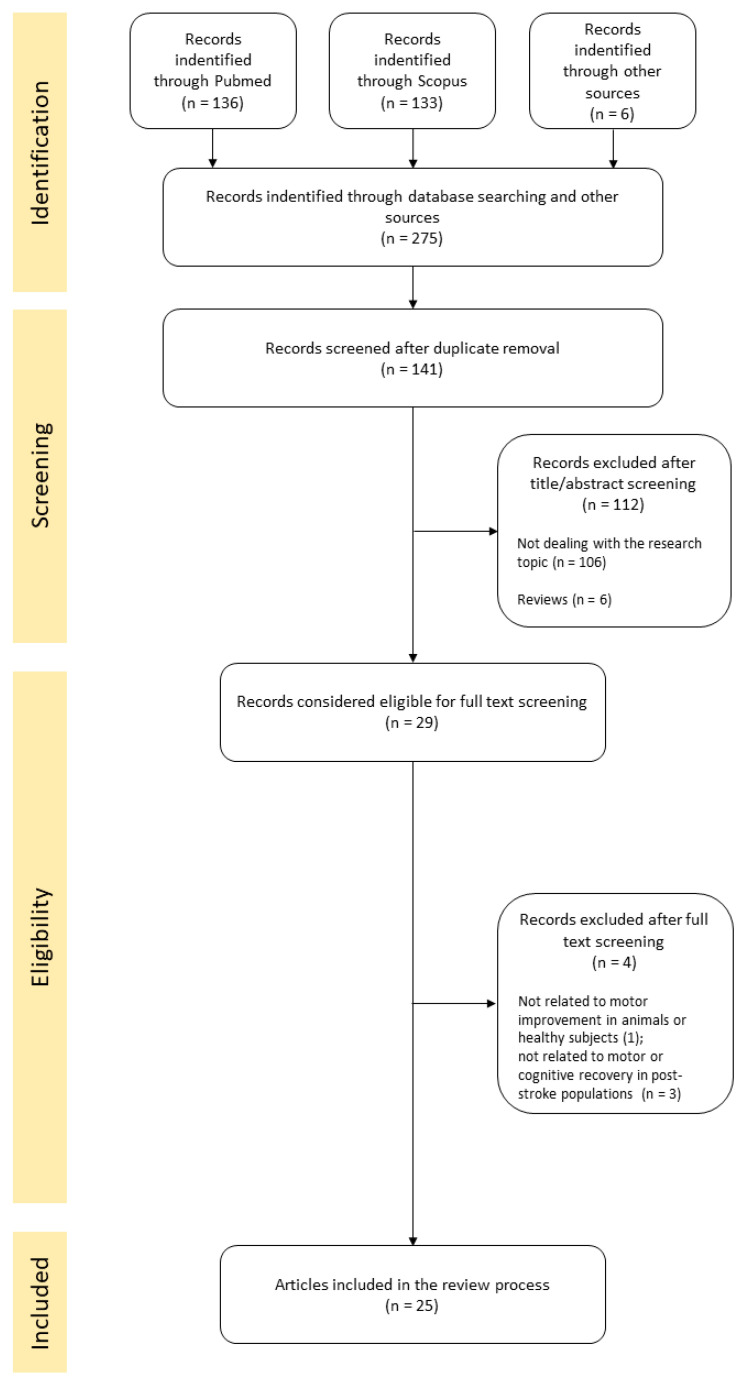
Flow diagram of review process.

**Figure 2 ijms-24-10173-f002:**
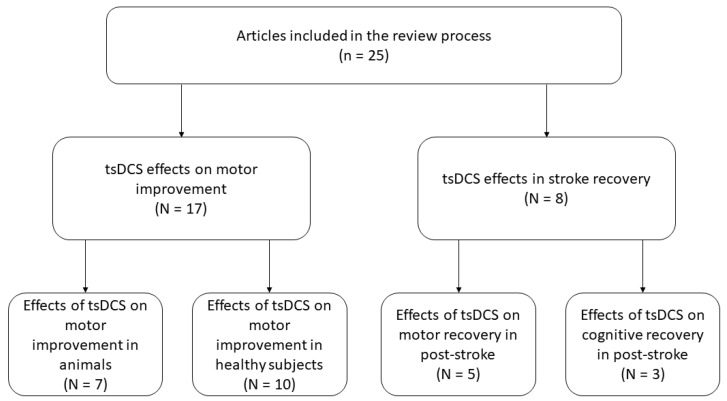
Subdivision of studies by topic.

**Table 1 ijms-24-10173-t001:** tsDCS studies on motor improvement in animals.

Articles	Populations	Target	Stimulation Polarity and Intensity	Duration and Number of Sessions	Results
**Animals**					
Ahmed, 2016 [38]	35 adult male mice (body weight, 40–55 g)	tsDCS electrode (10 mm wide and 15 mm long) over the T13-L6 vertebrae.	Anodal and cathodal, 0.5 mA	3 min of tsDCS during the duration of the reflex.	Motor neurons showed increased responses to cathodal and decreased responses to anodal tsDCS.
Bączyk et al., 2019 [39]	20 adult male rats (body weight 400–520 g)	A circle-shaped electrode (5 mm in diameter) on the lumbar vertebra. A metal clip placed on the abdominal skin flap ventrally to the lumbar spinal cord served as a reference electrode.	Anodal and cathodal, 0.1 mA	tsDCS was applied for 15 min, and motoneurons responses were intracellularly measured before, during, and after stimulation.	Anodal tsDCS potentiated motoneuron responses while cathodal tsDCS determined firing inhibition.
Bączyk et al., 2020 [40]	26 adult male rats (body weight 400–520 g)	A circle-shaped electrode (5 mm in diameter) on the lumbar vertebra. A metal clip placed on the abdominal skin flap ventral to the lumbar spinal cord served as a reference electrode.	Anodal and cathodal, 0.1 mA	tsDCS was applied for 15 min, and no recordings were made during that period.	Facilitatory changes were present only after anodal tsDCS and persisted for 30–60 min after stimulation.
Bączyk, et al., 2020 [41]	18 adult male rats (body weight: 384–450 g)	A rectangular-shaped electrode (5 × 10 mm) above the L1 lumbar vertebra, while a metal clip placed on the abdominal skin flap ventrally to the lumbar spinal cord served as a reference electrode.	Anodal, cathodal, and sham, 0.5 mA	tsDCS was applied for 15 min, 5 days per week for 5 weeks. Sham control group rats served as a reference.	Anodal tsDCS exerted facilitation of motoneuron responses, while cathodal effects were not significant.
Highlander et al., 2022 [42]	Three experimental groups with amyotrophic lateral sclerosis (ALS) of male transgenic mice	Electrodes were placed over the lumbar region of the spinal cord, with one electrode on the back and the other on the abdomen.	Anodal, cathodal, and sham, 0.5 mA	tsDCS was applied for 30 min for 16 daily treatments.	Only anodal tsDCS disrupted normal disease progression.
Song and Martin, 2022 [43]	5 adults male rats (body weight 280–320 g)	The active electrode was placed dorsally over C4 to T1 vertebrae, and the return electrode over the chest.	Anodal and cathodal, 1 mA	tsDCS was ramped over a 3s period to the maximal current, which was maintained for 20 s, and ramped back to zero during a 3 s period. Two sessions with 7 days between each session.	Both cathodal and anodal tsDCS immediately increased spontaneous motor unit firing during stimulation.
Williams et al., 2022 [44]	7 cats	The target electrode over C2–C6, and the return electrode on the sternal manubrium.	Cathodal and anodal, 1–5 mA	The duration of tsDCS was on for 40 s, with a 30 s ramp-up and a 30 s ramp-down period.	Cathodal/anodal current intensity modulated MEP enhancement/suppression, with higher cathodal sensitivity.

**Table 2 ijms-24-10173-t002:** tsDCS studies on motor improvement in humans.

Articles	Populations	Target	Stimulation Polarity and Intensity	Duration and Number of Sessions	Results
**Humans**					
Berry et al., 2017 [45]	12 (3 female) healthy volunteers (*M* ± *SD*: age 29 ± 11 years)	A pair of electrodes were placed over T11-T12 vertebrae, and a second pair was placed longitudinally on the abdomen.	Anodal and sham, 2.5 mA	Double-blind, randomized, crossover, sham-controlled design. 15 min of anodal tsDCS on repeated vertical countermovement jump (VCJ) performance at 0, 20, 60, and 180 min post-stimulation.	The magnitude and direction of change in VCJ performance was greater after anodal tsDCS than in the sham condition.
Sasada et al., 2017 [46]	15 healthy male volunteers	The target electrode was placed over T11 to L1 while the reference electrode was on the right shoulder.	Anodal, cathodal, and sham, 3 mA	15 min of tsDCS with ramping up and down for 15 s.	tsDCS improved sprint performance. The effect was larger for cathodal tsDCS than for anodal tsDCS.
Albuquerque et al., 2018 [47]	12 healthy volunteers (6 males; 24.75 ± 2.77 years)	The active electrode was placed over T11-T12, and the reference electrode was on right shoulder.	Anodal, cathodal, and sham, 2.5 mA	tsDCS was delivered for 1200s fade-in and fade-off 10 s. Sham stimulation followed the same montage of anodal stimulation, but after 30 s, the stimulator was turned off.	Anodal tsDCS/treadmill exercise reduced MEP’s amplitude and nociceptive flexion reflex (NFR) compared to the sham condition. Conversely, cathodal tsDCS/treadmill exercise increased NFR.
Awosika et al., 2019 [48]	43 healthy volunteers (24 women and 19 men; mean age ± SD, 25.9 ± 4.8 years)	The anode/sham electrodes were centered over T-11. The reference electrode was placed over the right shoulder.	Anodal and sham, 2.5 mA	Two groups underwent 20 min of backwards locomotion training (BLT) with concurrent anodal (n = 21) or sham (n = 22) tsDCS over three consecutive days.	Simultaneous application of anodal tsDCS with BLT facilitated the acquisition of locomotor skills.
Yamaguchi et al., 2020 [49]	4 experiments with different groups of healthy volunteers	The cathode was placed over T11-T12, and the reference electrode was on the right shoulder.	Cathodal and sham, 2.5 mA	Exp1: 10 min cathodal tsDCS. Exp2: Corticospinal excitability was examined by applying 15 single TMS pulses prior to and following tsDCS (2, 10, 20, and 30 min after). Exp3: 3 min cathodal tsDCS. Exp4: 10 min of cathodal stimulation + TMS as in Exp2.	Cathodal tsDCS facilitates voluntary motor output.
Zeng et al., 2020 [50]	Thirty R restless leg symptoms (RLS) subjects (23 females and seven males; mean age: 62.1 ± 8.04 years)	The anode was placed over T10, and the reference electrode was above the right shoulder.	Anodal and sham, 2 mA	tsDCS was delivered for 20 min. The treatment was applied daily for 14 consecutive days. In the sham condition, the stimulator was turned off after 30 s.	a-tsDCS improved the sleep and RLS symptoms in RLS patients.
Kamali et al., 2021 [51]	14 experienced male boxers	Anodal tDCS over the primary motor cortex (M1) and paraspinal region (corresponding to the hand area). Both cathodal electrodes were placed bilaterally adjacent to spinous processes of C5-T1(tsDCS).	Anodal and sham, 2 mA	Random sequential real or sham. Two sessions with a 72 h interval. 13 min of stimulation each session.	Anodal tDCS+ tsDCS vs. sham decreased the mean error scores by 47.5% in the selective attention task.
Clark et al., 2022 [52]	23 older adults (age = or >65)	The anode was placed over T11-T12. The two cathode electrodes were placed on each side of the umbilicus in approximately the same horizontal plane as the anode.	Anodal and sham, 2.5 mA	tsDCS was delivered for 30 min simultaneously with 15 trials of the complex terrain course involving stepping over foam obstacles and walking on compliant surfaces.	The anodal group showed greater performance than the sham group.
Fava De Lima et al., 2022 [53]	17 healthy volunteers	Three electric stimulation protocols were investigated: cathode over T10 and the reference electrode over the iliac crests; anode over T10 and the reference electrode over the iliac crests; sham.	Anodal, cathodal, and sham, 5 mA	tsDCS was delivered for 20 min. Measures of postural sway, both global and structural, were computed before, during, and following tsDCS period.	No significant changes were found after tsDCS in postural sway during quiet standing.
Kamali et al., 2023 [54]	15 experienced male taekwondo players	Anodal tDCS over the primary motor cortex (M1) and paraspinal region (corresponding to the hand area). Both cathodal electrodes were placed bilaterally adjacent to spinous processes of C5-T1 (tsDCS).	Anodal, cathodal, and sham, 2 mA	Two sessions of 13 min, 72 h apart. Next, the performance of the participants was evaluated through a simulation of taekwondo exercise directly after the sham and real sessions.	Anodal tDCS + cathodal tsDCS reduced reaction times in professional taekwondo practitioners.

**Table 3 ijms-24-10173-t003:** tsDCS studies on motor and cognitive recovery in post-stroke populations.

Articles	Populations	Target	Stimulation Polarity and Intensity	Duration and Number of Sessions	Results
**Motor Recovery**					
Picelli et al., 2015 [55]	30 chronic stroke patients with mild–severe residual walking impairment	tDCS anode over the ipsilesional primary motor area and the cathode above the contralateral orbit. tsDCS cathode was placed over T9-T11, and the anode was above the shoulder of the unaffected hemibody.	Anodal, cathodal, and sham, 2.5 mA	10–20 min of robot-assisted gait training sessions, five days a week, for 2 consecutive weeks combined with anodal tDCS + sham tsDCS (group 1) or sham tDCS + cathodal tsDCS (group 2) or anodal tDCS + cathodal tsDCS (group 3). The primary outcome was the 6 min walk test (6MWT) performed before, after 2 and 4 weeks post-treatment.	Significant differences in the 6MWT were noted between groups 3 and 1 at post-treatment and at 2-week follow-ups and between group 3 and group 2. No difference was found between group 2 and group 1.
Picelli et al., 2018 [56]	20 chronic stroke patients with mild–severe residual walking impairment	tDCS anode over the ipsilesional primary motor area and the cathode above the contralateral orbit. tsDCS cathode was placed over T9-T11, and the anode was placed above the shoulder of the unaffected hemibody.	Anodal and cathodal, 2.5 mA	10–20 min of robot-assisted gait training sessions, 5 days a week, for two consecutive weeks. Group 1 underwent online cathodal tDCS over the contralesional cerebellar hemisphere + cathodal tsDCS. Group 2 received online anodal tDCS over the ipsilesional hemisphere + cathodal tsDCS.	Cathodal tDCS over the contralesional cerebellar hemisphere + cathodal tsDCS boosts the effects of robot-assisted gait training in chronic stroke patients with walking impairment.
Picelli et al., 2019 [57]	40 chronic stroke patients with mild–severe residual walking impairment	tsDCS cathode over the cerebellar hemisphere and the anode over the buccinator muscle on the same side. tsDCS cathode was placed over T9-T11, and the anode was above the shoulder of the unaffected hemibody.	Cathodal, 2 mA	10–20 min of robotic gait training sessions, five days a week, for two consecutive weeks. Two groups: cathodal tDCS over the contralesional cerebellar hemisphere + cathodal tsDCS in combination with robotic training; cathodal tDCS over the ipsilesional cerebellar hemisphere + cathodal tsDCS in combination with robotic training.	Cathodal cerebellar tDCS over the contralesional or ipsilesional hemisphere + cathodal tsDCS led to similar effects in robotic gait training.
Paget-Blanc et al., 2019 [58]	26 chronic stroke patients with upper limb spasticity	The anode was placed on the C6 spine level, and the reference electrode was above the iliac crest on the abdomen.	Anodal and sham, 2.5 mA to 4 mA	Patients received five consecutive daily sessions of 20 min of anodal tsDCS or sham + pDCS (peripheral nerve direct current stimulation). Each session was separated by one week of washout period.	Anodal tsDCS + pDCS significantly reduced upper limb spasticity in participants with stroke. Decreased spasticity was persistent for five weeks after treatment and was accompanied by improved motor function.
Awosika et al., 2020 [59]	30 chronic stroke patients with mild–severe residual walking impairment	The anode/sham over T11 and the reference electrode over the right shoulder.	Anodal and sham, 2.5 mA	6–30 min sessions (three sessions/week) of backward locomotor treadmill training, with concurrent anodal or sham tsDCS. Sham tsDCS was delivered over a period of 30 s at the beginning and end of the stimulation period.	Anodal tsDCS did not enhance the degree of improvement in walking speed and capacity, relative to backward locomotor treadmill training + sham.
**Cognitive Recovery**					
Marangolo et al., 2017 [33]	14 chronic post-stroke aphasics	Anode on T10-T11 and the reference electrode over the right shoulder of the deltoid muscle.	Anodal, cathodal, and sham, 2 mA	20 min of tsDCS during a verb- and noun-naming task. Each experimental condition was run in five consecutive daily sessions over 2 weeks.	A significant improvement was found only in verb naming after anodal tsDCS with respect to the other two conditions, which persisted at 1 week after the end of the treatment.
Marangolo et al., 2020 [34]	16 chronic post-stroke aphasics	Anode on T10-T11 and the reference electrode over the right shoulder of the deltoid muscle.	Anodal and sham, 2 mA	20 min of stimulation during a verb-naming task. Each experimental condition was run in five consecutive daily sessions over two weeks.	After anodal tsDCS, a significant improvement was found only in verb naming, which positively correlated with connectivity changes in a cerebellar–cortical network.
Pisano et al., 2021 [35]	10 chronic post-stroke aphasics	Anode on the T10-T11 and the reference electrode over the right shoulder of the deltoid muscle.	Anodal and sham, 2 mA	Five days of tsDCS with a concomitant repetition training for articulatory deficits.	Only after anodal tsDCS did patients exhibit better accuracy in repeating the treated items. These effects persisted at F/U and generalized to other oral language tasks.

## Data Availability

Data sharing not applicable.

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
