# Peer review of "What Else Can Be Done by the Spinal Cord? A Review on the Effectiveness of Transpinal Direct Current Stimulation (tsDCS) in Stroke Recovery"

_ijms, 2023, doi:10.3390/ijms241210173_

Round 1
Reviewer 1 Report
May 20, 2023
Review of the article “What Else Can Be Done by the Spinal Cord? A Review on the Effectiveness of Transpinal Direct Current Stimulation (tsDCS) in Stroke Recovery” submitted to the International Journal of Molecular Sciences
This is a very interesting scoping review of new methods that may be applied in the treatment of people affected by stroke. The authors cite a significant amount of methodological data that are important in analyzing the obtained results, but this can also create new questions. For example, the authors pay much attention to the polarity of applied stimulations, showing that the results obtained do depend on the polarity. It might not be surprising as there are different thresholds of electrical stimulation depending on the polarity[1] or directions of affected pathways. It would help if the authors explained more the meaning (if known) of cited results as a function of polarity. They should cite a relevant study.
Focusing on recovery from a stroke is a helpful limitation for the review, but the authors should consider a paragraph concerning the general description of neuronal plasticity evoked by electrical stimulation.
360 -361 Since the authors conclude that “It seems, therefore, likely that, to exert an effect over time on motor recovery following a stroke, either cathodal or anodal tsDCS should be combined with cortical [76–78] or peripheral stimulation [79]”, it may be appropriate to mention that there are other studies that use peripheral electrical stimulation of the vagus nerve.[2]
[1] Anna Dalal Kirsch, Sharon Hassin-Baer, Cordula Matthies, Jens Volkmann, Frank Steigerwald. Anodic versus cathodic neurostimulation of the subthalamic nucleus: A randomized-controlled study of acute clinical effects. Parkinsonism & Related Disorders, 55, 2018, 61-67. https://doi.org/10.1016/j.parkreldis.2018.05.015
[2] Meyers EC, Solorzano BR, James J, Ganzer PD, Lai ES, Rennaker RL 2nd, Kilgard MP, Hays SA. Vagus Nerve Stimulation Enhances Stable Plasticity and Generalization of Stroke Recovery. Stroke. 2018 Mar;49(3):710-717. doi: 10.1161/STROKEAHA.117.019202
Author Response
- It would help if the authors explained more the meaning (if known) of cited results as a function of polarity.
We totally agree with the Referee’s request that we have omitted to clearly explain the results obtained by the different tsDCS studies as a function of polarity. Accordingly, in the revised version, we have reported all the explanations given by the authors with reference to the different electrode montages used.
- Focusing on recovery from a stroke is a helpful limitation for the review, but the authors should consider a paragraph concerning the general description of neuronal plasticity evoked by electrical stimulation.
Our review was principally focused on stroke recovery since this request was made by the Journal Guest Editor in order to fully correspond to the topic of his Special Issue in which the review will appear. However, we agree with the Referee’s criticism that our review lacked of an explanation on the neurobiological mechanisms responsible for neuroplastic changes due to tsDCS. Accordingly, in the Discussion we have added a paragraph which explain the neurobiological changes which are thought to be elicited by tsDCS.
- Since the authors conclude that “It seems, therefore, likely that, to exert an effect over time on motor recovery following a stroke, either cathodal or anodal tsDCS should be combined with cortical [76–78] or peripheral stimulation [79]”, it may be appropriate to mention that there are other studies that use peripheral electrical stimulation of the vagus nerve.[1][2.
We thank the Referee for his/her suggestion. In the revised version, in the discussion, we have briefly mentioned a couple of studies which showed that the enhancement of corticospinal network plasticity to increase synaptic connectivity for motor rehabilitation after stroke has also been reported, both in humans and animals, by using vagus nerve stimulation (VNS) in conjunction with rehabilitative training.
Reviewer 2 Report
This is a very interesting review of the role of transpinal direct current stimulation in stroke recovery and motor activity in healthy animals and patients. I appreciate the enormous load of work from the reviewers in the attempt to review several different aspects of tsDCS and how challenging it is to summarize all data in a manuscript. But because of the amount of information provided, it is challenging for the reader to fully comprehend the potential of the review and what are the gaps and conclusions of the studies reviewed. Please find the specific concerns raised below:
1) The introduction section is very interesting, but very long. It presents itself as a narrative review at first. Then, the search strategy is added, and a new set of information is provided. For example, the “tsDCS and plasticity” and “neurophysiology techniques and tsDCS studies” could be considered a narrative review by itself or even a discussion to corroborate the role of tsDCS in the improvement of stroke rehabilitation.
2) I strongly suggest separating in different tables the animal studies from the healthy human data. That would help the presentation of the table and would also allow for a better discussion of the mechanism’s studies after tsDCS. Please consider that the electrode and current ratio in small animals is very different from humans (which is also true for classic tDCS), which may lead to misinterpretation of data.
3) The concern raised above may also aid in a better comprehensive review of the data presented in the table. There is too much text information in each column, making it hard for the reader to fully comprehend what is being reviewed. What is the difference, in terms of efficacy, from anodal to cathodal stimulation? Is there a difference between combining tDCS with tsDCS? These questions could be better discussed in the review. A possibility is to divide the discussion into protocols, efficacy, etc.
4) I do appreciate that protocols are very distinct from one another, but I suggest that the authors discussed a little bit more about what we still do not know about tsDCS, and how can we improve tsDCS clinical trials going forward.
Author Response
- The introduction section is very interesting, but very long. It presents itself as a narrative review at first. Then, the search strategy is added, and a new set of information is provided. For example, the “tsDCS and plasticity” and “neurophysiology techniques and tsDCS studies” could be considered a narrative review by itself or even a discussion to corroborate the role of tsDCS in the improvement of stroke rehabilitation.
We thank the Referee for his/her corteous and helpful comments. Accordingly, we have shortened the Introduction moving some paragraphs in the discussion. Following also Referee’s 1 suggestion, that our review lacked of an explanation on the neurobiological mechanisms responsible for neuroplastic changes due to tsDCS, in the Discussion we have added a paragraph which explain the neurobiological changes which are thought to be elicited by tsDCS.
- I strongly suggest separating in different tables the animal studies from the healthy human data. That would help the presentation of the table and would also allow for a better discussion of the mechanism’s studies after tsDCS.
We agree with the Referee’s request. Accordingly, we have considered two different tables (1 and 2) for animals and healthy humans studies.
- The concern raised above may also aid a better comprehensive review of the data presented in the table. There is too much text information in each column, making it hard for the reader to fully comprehend what is being reviewed. What is the difference, in terms of efficacy, from anodal to cathodal stimulation? Is there a difference between combining tDCS with tsDCS? These questions could be better discussed in the review. A possibility is to divide the discussion into protocols, efficacy, etc.
We totally agree with the Referee’s request. Indeed, in the revised version, we have reported all the explanations given by the authors with reference to the different polarities and electrode montages used.
- I do appreciate that protocols are very distinct from one another, but I suggest that the authors discussed a little bit more about what we still do not know about tsDCS, and how can we improve tsDCS clinical trials going forward.
We agree with the Referee’s request that our review lacked of an explanation on what is still not known about tsDCS. Accordingly, in the last part of the Discussion, we have added a paragraph which highlights the need to further investigate several issues regarding the use of tsDCS to enhance its efficacy to plan the best rehabilitation protocols.
Round 2
Reviewer 2 Report
The authors have successfully answered all my concerns.